

# Spatio-Temporal Patterns of Accumulation and Surface Roughness in Interior Greenland with a GNSS-IR Network

Derek J. Pickell[1], Robert L. Hawley[1], and Adam LeWinter[2]

[1]Department of Earth Sciences, Dartmouth College, Hanover, New Hampshire, USA
[2]Cold Regions Research and Engineering Laboratory, Hanover, New Hampshire, USA

**Correspondence:** Derek J. Pickell (derek.gr@dartmouth.edu)

**Abstract.** The dry snow zone is the largest region of the Greenland Ice Sheet, yet temporally and spatially dense observations of surface accumulation and surface roughness in this area are lacking. We use the global navigation satellite system (GNSS) interferometric reflectometry (GNSS-IR) technique with a novel, low-cost GNSS network of twelve stations in the vicinity of the ice sheet summit to reveal temporal and spatial patterns of accumulation of the upper snow layer. We show that individual

measurements are highly precise ($\pm 2.8$ cm), while the aggregate of hundreds of daily measurements across a large spatial footprint can detect millimeter-level surface changes and are biased by -2.7 $\pm$ 3.0 cm compared to a unique validation dataset that covers a similar spatial extent as the instrument sensing footprint. Using the validation dataset, we find that the reflectometry technique is most sensitive to the surrounding 4–20 m of the surface, with the antenna at a height of 1–2 m above ground level. Along with an exceptionally high rate of accumulation at the beginning of the study, we also detect an across-slope dependence

in accumulation rates at yearly timescales. For the first time, we also validate GNSS-IR sensitivity to meter-scale surface heterogeneities such as sastrugi, and we construct a time series of seasonal surface roughness evolution that may hint at a seasonal pattern of heightened wintertime roughness features in this region. These surface accumulation and roughness measurements provide a novel dataset for these critical variables and show a statistically significant (p < 0.01) relationship with occurrences of both high winds and precipitation events, but only moderate correlations, suggesting other processes must also contribute to

accumulation and enhanced surface roughness in the interior region of Greenland.

## 1 Introduction

The surface mass balance (SMB) of the Greenland Ice Sheet is a key indicator of its response to a dynamic climate; with an acceleration of summertime melt and overall surface mass loss, sea levels are directly impacted (van den Broeke et al., 2016; Smith et al., 2020). In central Greenland, SMB is still largely positive but the patterns of accumulation in this region play a

critical role in the stability and the evolution of the ice sheet (e.g., McConnell et al., 2000). Quantifying accumulation and its meter-scale variability (closely linked to surface roughness) is an important part of tracking SMB, and an understanding of these variables is also important for assessing stratigraphic noise in ice core interpretations (van der Veen and Bolzan, 1999), determining turbulent heat fluxes over the ice sheet (van Tiggelen et al., 2021), modeling firn gas exchange (Albert and Hawley, 2002), and validating space-borne radar (Scanlan et al., 2023). Meanwhile, the drivers of accumulation and surface roughness



in the dry snow zone, such as precipitation, sublimation, and wind erosion, are hard to measure given temporally and spatially sparse ground-based observations, complicating the formulation of a process-level understanding of accumulation (Castellani et al., 2015).

The few long-term measurements of accumulation or surface roughness in Central Greenland that are made commonly sacrifice some aspect of spatial or temporal scale: for example, ice cores provide a record of accumulation and climatic patterns

over long timescales but no present-day information. On their own, cores are point measurements in space and therefore reveal little in terms of locally varying processes such as wind erosion (Kuhns et al., 1997). As such, factors such as spatially and seasonally varying accumulation must be understood in order to quantify the stratigraphic noise in ice cores (Fisher et al., 1985). To better interpret ice core records and connect them with modern surface processes, near-real-time studies have been established on the ice sheets: for example, stake fields in the vicinity of coring sites can better capture the modern

spatial-varying accumulation signal, with measurements often made on weekly or yearly scales (e.g., Dibb and Fahnestock, 2004). However, snow stakes are manual measurements and therefore pose logistical constraints, while approximations must be made to convert surface height measurements to accumulated water equivalent volume (e.g., Takahashi and Kameda, 2007). A more recent method that has attempted to address this gap uses a buried cosmic ray counter to directly quantify surface mass balance (Howat et al., 2018). While this method better addresses the aforementioned shortcoming with the snow stake

method in converting heights to mass, this technique may still be limited in its spatial footprint. Lastly, remotely sensed surface height change, such as with the ICESat-2 laser altimeter, can provide ice sheet-wide coverage, while temporal gaps and its spatial resolution make studying small-scale surface processes difficult (van Tiggelen et al., 2021). As such, there remains an opportunity in providing measurements at higher temporal resolution and duration, along with greater spatial representativeness for a given area of study.

Here, we leverage a network of precise, easily deployable, low-cost global navigation satellite system (GNSS) instruments that were positioned for two years over tens of kilometers to expand both spatial and temporal measurements of surface accumulation and surface roughness. To retrieve surface measurements, we employ GNSS interferometric reflectometry (GNSS-IR), whereby the direct and reflected signals from GNSS satellites incident upon the GNSS receiver antenna create a characteristic signal-to-noise (SNR) interference pattern that directly corresponds to the antenna height above the snow surface (Larson et al.,

2009). Unlike point measurements of the snow, GNSS-IR can sense a large area ($\sim$100's to 1000's m$^2$) due to the azimuthal distribution of reflected GNSS signals about the instrument, and this method can produce a temporally dense or even continuous series of surface measurements depending on the logging configuration of the receiver. This technique has been used in other studies to measure accumulation throughout the cryosphere, such as in alpine environments (Larson, 2016), on the Greenland Ice Sheet (Larson et al., 2020; Dahl-Jensen et al., 2022), and in Antarctica (Siegfried et al., 2017; Pinat et al., 2021).

With our GNSS network, we not only examine the time series of the 24 h mean surface height at each station, but also the surface height variability at each individual station and between each station. While these aforementioned GNSS-IR studies produce singular height averages during a certain time period, we extend the GNSS-IR technique by evaluating the spatial heterogeneities (roughness) within the instrument sensing footprint. The time series of surface roughness addresses a critical gap in cryosphere observations of small-scale surface roughness evolution through time. Together with accumulation mea-



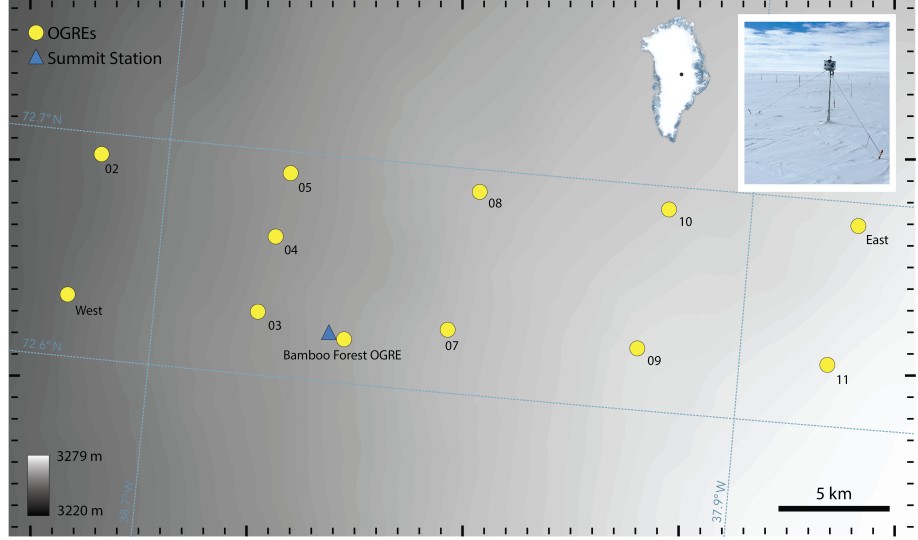

**Figure 1.** Open GNSS Research Equipment (OGRE) array in the Summit vicinity. Inset: example OGRE setup with the GNSS antenna, instrument, and solar panel mounted on a pole $\sim 2$ m above ground, and with the battery buried just below the surface. This particular instrument was placed in the Summit Station Bamboo Forest, an 11 by 11 stake array covering a similar area to the OGRE sensing footprint. The background shading highlights the elevation slope from the ice divide to the west, and is derived from MEaSUREs Greenland Ice Mapping Project 2 (Howat et al., 2014).

surements, we can take advantage of the temporal and spatial resolution of these data to link patterns of accumulation and roughness with precipitation and winds.

This paper is organized as follows: first, we provide information on the GNSS network, instrumentation, and auxiliary data, along with background on the GNSS-IR technique and its spatial properties. Next, we statistically validate the GNSS-IR technique by comparing surface height measurements with a long-running snow stake field of similar spatial extent, being

the first study to our knowledge to assess GNSS-IR-derived measurements with those made over a comparable footprint in the cryosphere. We also extend this analysis by determining the precision of individual GNSS-IR measurements and evaluating any differences between L1 and L2 GNSS frequencies. Then, we assess the GNSS-IR technique in quantifying surface roughness that arises from heterogeneities in local accumulation. Finally, we analyze the daily, seasonal, and spatial patterns of accumulation and surface roughness derived from this technique, and connect these measurements to occurrences of high

winds and precipitation.



## 2 Instruments and Methods

### 2.1 Open GNSS Research Equipment (OGRE) Network

The network of twelve GNSS stations spans a 35 km east-west transect in the Summit Station vicinity of the Greenland Ice Sheet, with the easternmost stations positioned near the ice divide (Fig. 1). These stations were originally deployed to record surface velocity, validate ICESat-2 laser altimetry height estimates of the ice sheet, and demonstrate the utility of low-cost, high-precision instruments in the cryosphere, and here we use them to analyze accumulation patterns in the network area using the GNSS-IR technique. Each station is built from a low-cost, low-power, multi-GNSS instrument called an OGRE, designed specifically for rapid, overwinter deployments (Pickell and Hawley, 2024b). Station configuration includes a lightweight patch antenna mounted on a 3 m pole, a 10 W solar panel with the instrument mounted on the backside, and a 40 Ah battery buried below the surface to minimize drifting. Most stations recorded 1 Hz data for 24 hr periods once, twice, or four times monthly, year round, to coincide with ICESat-2 overpasses, and the Bamboo Forest OGRE recorded 24 hr data once weekly. Due to a chip shortage during the fabrication of these instruments, three stations (West, 07, 09) were built with a sister chip that tracks GNSS satellites at L1 and L5 frequencies instead of L1 and L2; we do not analyze the L5 signals in this paper.

### 2.2 MERRA-2 Atmospheric Reanalysis Product

We use NASA's Modern-Era Retrospective analysis for Research and Applications, Version 2 (MERRA-2) for the synoptic variables in this study (Global Modeling And Assimilation Office and Pawson, 2015). Due to the intermittency of meteorological data during the winter of 2022 at Summit Station, we chose to use MERRA-2 for the timing and magnitude of wind and precipitation events as part of our surface process analysis. MERRA-2 and other reanalysis products provide reliable near-surface climatic conditions (e.g., Wang et al., 2019; Gossart et al., 2019), but in ice sheet environments, biases may still exist: MERRA-2 is shown to underestimate precipitation and evaporative processes that lead to surface accumulation (Siegfried et al., 2017; Howat, 2022). MERRA-2 variables are linearly interpolated to each of the OGRE station locations.

### 2.3 Snow Stake Array

The snow stake array, also referred to as the "Bamboo Forest," is arranged in its current configuration into an 11 by 11 stake grid with each stake spaced approximately 8 m apart (80 m by 80 m). The array is located east of Summit Station in a region of relatively undisturbed snow, and the grid is rotated approximately 15° clockwise from true north to align with the prevailing winds (Fig. 2a). Measurements are made on a weekly basis when weather conditions allow, and data have been collected since 2003. Heights from a fiducial mark on each stake are made to the snow surface, and are sensitive to the same snow surface height change as measured by the OGRE stations with GNSS-IR. One of the OGREs is located within the Bamboo Forest ("Bamboo Forest OGRE") and programmed to log once weekly on Wednesdays, the target day of the manual Bamboo Forest survey.





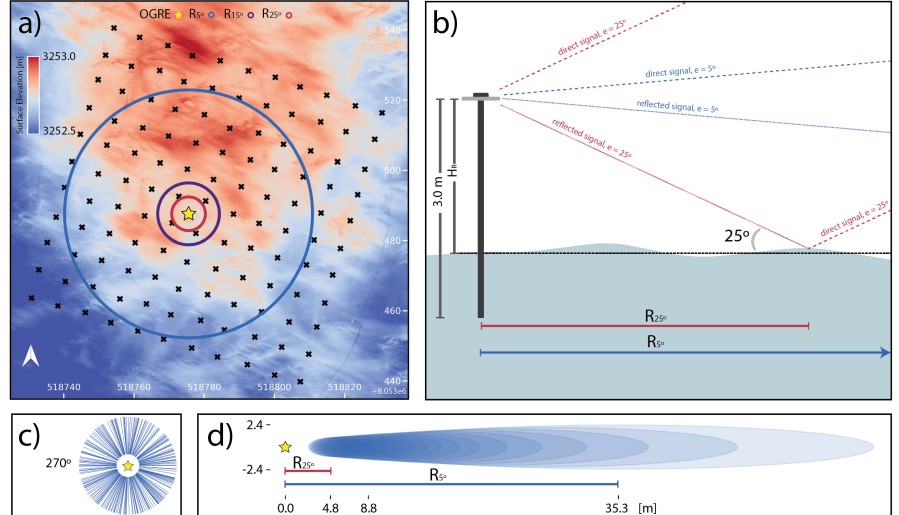

**Figure 2.** (a) Plan view of Bamboo Forest OGRE, projected into WGS 84 / UTM zone 24N. The reflection points corresponding to satellite elevations of 5°, 15° and 25° are drawn radially about the OGRE (calculated based on a 2 m antenna height) in blue, purple and red respectively. This OGRE is located in the Bamboo Forest, a 121 accumulation stake network which we used to validate GNSS-IR. The background is the TLS-derived surface elevation from 16 June 2024. (b) Elevation view of OGRE antenna and GNSS-IR reflection geometry, showing the antenna mounted on a partially buried 3.0 m pole (instrument not shown). As a GNSS satellite rises (or sets) through the elevation range of 5° to 25°, the location of the surface reflection point changes from a maximum radial distance from the antenna at 5° to a minimum distance at 25°. (c) Plan view of the locations of all observed satellite arcs from the perspective of the OGRE over a 24 h period on August 18, 2022. (d) Detailed plan view of the Fresnel zone geometry of a single satellite arc from Panel C relative to the OGRE, with the reflecting signal ellipse center point labeled at satellite elevation angles corresponding to R = 5° and R = 25°.

As each stake height measurement is unreferenced to all others, we surveyed the Bamboo Forest with an auto level on the 14th of June, 2024 to reference all stakes and the Bamboo Forest OGRE to the same horizontal reference datum. This allows for the direct comparison of the biases between the Bamboo Forest stakes and the Bamboo Forest OGRE. The snow surface in the Bamboo Forest was also scanned with a VZ-2000i Terrestrial Laser Scanner (TLS) on the 16th of June, 2024 for further

inter-comparison and spatial analysis of the OGRE data at 10 cm horizontal spatial resolution (Fig. 2a).

## 2.4    Surface Height Change and GNSS-IR Technique

Surface height is the distance from the antenna phase center on the OGRE or the fiducial mark on the bamboo stakes to the snow surface. In GNSS-IR literature, this value is commonly called the reflector height ($H_r$) to indicate that the measurement is the vertical height from the reflecting surface (snow-air interface) to the antenna phase center. We adopt this nomenclature

when referring to the OGRE antenna heights above the surface. Changes in $H_r$ are sensitive to several processes in the dry



snow zone (Castellani et al., 2015):

$$\Delta H_r = P + M + L + C + W \tag{1}$$

Where $P$ is precipitation as snowfall, $M$ is melt (generally negligible at Summit), and $L$ is surface change due to latent heat flux, and can be positive or negative if in a depositional regime or a sublimation regime, respectively, depending on atmospheric
conditions. $C$ is compaction between the surface and the bottom of the pole, and we assume the pole to be locked into the layer at its base per Takahashi and Kameda (2007). Finally, $W$ is wind redistribution, which can also be positive (depositional) or negative (erosional). In this paper we also refer to $\Delta H_r$ as "accumulation", which can be positive or negative and represents the change in the snow layer between the measured surface and the anchored bottom of the pole.

In calculating $H_r$, GNSS-IR does not rely on any position-related measurements made by the instrument; instead, the only
data of interest are the SNR levels measured by the instrument for each satellite signal. When tracking satellites at low elevation angles, the gain characteristics of a zenith-pointing GNSS antenna are such that they are not only susceptible to the direct incoming satellite signal, but also any reflected signal, called multipath (Fig. 2b). At the antenna phase center, these signals interfere, and whether they do so constructively or destructively at a given satellite elevation angle is a function of antenna height above the reflecting surface. Specifically, the frequency of the interference pattern as the satellite rises is directly related
to the ratio of the antenna height and the signal wavelength (Georgiadou and Kleusberg, 1988). With multiple L1 and L2 constellations, we can use this technique to derive hundreds of $H_r$ estimates each day, distributed radially about the instrument based on the azimuth directions of the satellites observed at the latitude of the instrument (Fig. 2c). In the ice sheet interior, this technique is particularly effective given the unambiguous, near-planar reflecting surface.

Following Roesler and Larson (2018), our processing strategy involves masking satellite arcs from 5°–25° (the elevation
range for which we observe strong multipath in our receivers), fitting a 4th order polynomial to remove the direct signal trend, correcting for refraction with a simple bending model, and estimating the interferometric frequency with a Lomb-Scargle periodogram. Successful Lomb-Scargle $H_r$ estimates are filtered by a peak-to-noise ratio that ensures the detected $H_r$ estimate is the dominant signal by a factor of 4.0. The software used to process these data is freely available at https://doi.org/10.5281/zenodo.10644225 (Larson, 2024).

In contrast to sonic snow depth sensors or snow stakes, the spatial footprint of GNSS-IR is complicated by the changing antenna height above the surface and the reflection patterns of each satellite arc, but this also provides an opportunity to sense a much larger area than traditional methods (e.g., Roesler and Larson, 2018). At our lower elevation angle bound of 5°, the reflection point is approximately 35 m away from the antenna at the beginning of our study, when the antenna pole is approximately 2.0 m above the surface. By the end of the study, when the antenna is closer to the surface by 1.0–1.5 m due
to accumulation, the outer reflection point has migrated inwards by about 10 m radially. Meanwhile, the inner reflection point when a given satellite has risen to 25° starts at 4.8 m for an antenna pole height of 2.0 m and migrates inwards around 2 m by the end of the study. The sensing footprint of a single $H_r$ measurement can be broadly characterized as an evolving ellipse centered somewhere on the reflection circle depending on the satellite azimuth, which becomes smaller as it migrates towards



the instrument as the satellite rises (Hristov, 2000; Larson and Nievinski, 2013) (Fig. 2d). These individual footprints then
combine into a near-circular footprint over a 24 h period of aggregated $H_r$ estimates.

For this study, we model the footprints as ellipses that change radially, but in reality, as a satellite rises or sets, its path across
the sky is not necessarily orthogonal to the pointing vector from the instrument to the satellite, and therefore the azimuthal
location may change slightly as the satellite rises or sets (Appendix A). This effect is small so we chose to use the mean
azimuth of each satellite across its 5°–25° arc, noting that this does induce an additional error when we estimate the ground
location of each reflection.

## 3 Surface Height Measurement Validation

### 3.1 Previous Validation Studies in the Cryosphere

Several studies have evaluated the performance of GNSS-IR in snowy environments. In a mountain saddle, Gutmann et al.
(2012) found a 10 cm bias between the receiver and a scanning laser rangefinder, with this variability largely attributed to the
surface noise recorded by the laser scanner. Siegfried et al. (2017) demonstrated that GNSS-IR reflections from an Antarctic
network of GPS stations overestimated $H_r$ compared to manual measurements by $2.0 \pm 6.0$ cm. In Greenland, closer to the
margin with sloping terrain, Dahl-Jensen et al. (2022) derived a RMSD of 17 cm and correlation of .98 compared to a sonic
ranger. Finally, Larson et al. (2020) determined a 9.9 cm standard deviation (no bias provided) of the differences between a
GPS station and a nearby ultrasonic sensor in the interior of Greenland, and points out that these uncertainties are within the
scale of snow roughness features that could bias smaller representative measurements such as those made by an ultrasonic
sensor when a sastrugi migrates within the field of view.

No study to our knowledge has attempted to reconcile GNSS-IR-derived $H_r$ estimates with their constitutive footprint within
the cryosphere; one study in the continental USA made measurements with consideration to the footprint of the GNSS reflected
signals, which found a –5.7 cm bias and 10.3 cm RMSE in snow depth estimates measured relative to the ground datum using
a snow probe (McCreight et al., 2014).

### 3.2 Evaluation of Bias and Precision Relative to Bamboo Forest Stake Network

In this study, we compare data with an OGRE placed in the center of the Summit Station Bamboo Forest. For the manual
bamboo measurements, each stake is referenced to the datum plane that was surveyed on 16 June 2024 with an auto level. This
survey allows us to add or subtract a correction factor to each bamboo stake at any point in time to reference it to a horizontal
plane. We estimate these correction factors have a maximum error of approximately 1.0 cm, based on the legibility of the stadia
rod from the furthest measuring location of the auto level. Throughout the year, stake heights are measured to the nearest half
cm, so we estimate an additional measurement error of 0.25 cm for each stake. Furthermore, we assume that the bamboo stakes
are all locked into the firn at their base at the same level, which needs to be true to continue to reference the stakes to the
horizontal plane backwards in time. We believe this is a reasonable assumption as the stakes are pushed into the snow until a



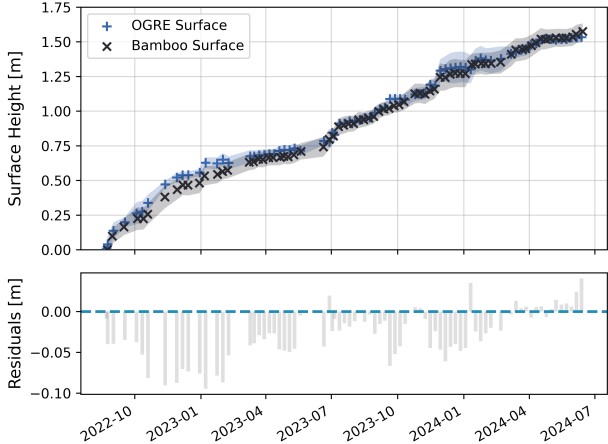

**Figure 3.** The comparison between the mean 24 h height estimates of the OGRE and the mean height estimate from a selection of the nearby 36 stakes within Bamboo Forest. Note that the $H_r$ estimates for both the OGRE and the stakes have been flipped to show the increasing surface height, as opposed to the decreasing height between the fiducial and the surface. The shaded uncertainties are taken to be the 1-$\sigma$ values of the OGRE height estimates for a 24 h period and the 1-$\sigma$ values of the heights of the stakes, corrected to the same datum.

firm layer is reached upon installation. Finally, as this study spans from August 2022 to June 2024, it encompasses an annual stake raise in June 2023, whereby the stakes were measured, raised in the firn and measured again. This may induce another small error based on the calculated distance between the pre-raised and post-raised stakes, which would be the error estimate for a given height measurement added with itself in quadrature, or 0.33 cm. Thus, there is a possibility for compounding errors up to 1 to 2 cm, plus any unquantified or correlated errors, retrospectively from the original auto level measurements. In other

words, the Bamboo Forest data recorded closer to the beginning of the study and corrected to the auto level datum are more uncertain.

    The Bamboo Forest OGRE records data for 24 h each Wednesday, but due to safety and weather conditions, the snow stakes were not always measured on this day. Thus, when comparing the two datasets, we define a ±48 h window relative to the OGRE logging day to search for a comparable stake survey. A prior study found the mean accumulation in the Bamboo Forest

to be 71 cm yr$^{-1}$ (Castellani et al., 2015), which corresponds to a mean daily signal of 1.9 mm. Thus, we estimate a several-mm additional error in our comparisons due to the temporal offset between datasets, which may be exaggerated during periods of high accumulation. To select the bamboo stakes that best represent the same sensing footprint of the OGRE, we iteratively average "concentric" stakes radially about the OGRE, starting with the closest four stakes (2 by 2 stake square) and up to five stakes away (10 by 10 stake square), while filtering any stake measurements > 3$\sigma$ from the mean of the entire network. We find

that the variance between the two datasets begins to increase after including the 3rd ring of stakes (6 by 6 stake square), thus we will only consider the 36 stakes that form a box around the OGRE.

    Overall, we observe a mean bias of -2.7 ± 3.0 cm (n = 76) between the averaged daily OGRE $H_r$ and the 6 x 6 stake array, with a correlation coefficient of 0.998 (r$^2$ = 0.996) (Fig. 3). Compared to the 6.0 cm variance found by Siegfried et al. (2017),



**Table 1.** Pairwise biases between L1 and L2 frequencies, showing very low biases for intra-constellation differences in L1 and L2 frequency $H_r$ estimates.

| Constellation | L1 Center Frequency (MHz) | L2 Center Frequency (MHz) | Mean Bias (m) | 1-SD (m) | n (pairs) |
| --- | --- | --- | --- | --- | --- |
| GPS | 1575.42 (C/A) | 1227.6 (L2C) | -0.0007 | 0.046 | 6647 |
| GLONASS | ~1602 (L1OF) | ~1246 (L2OF) | 0.0005 | 0.046 | 6643 |
| GALILEO | 1575.42 (E1-B/C) | 1207.14 (E5B) | -0.0015 | 0.046 | 4559 |

an F-test for equality of variances indicates that there is a significant large difference between the precisions ($p = 1.4 \times 10^{-9}$).

We attribute the higher precision in this study to the centered, spatial representativeness of the validation dataset, and perhaps a smoother reflecting surface. However, the OGRE underestimates $H_r$, especially in wintertime months, and this may be due to radio shadowing from surface features or a more difficult measurement environment due to snow build up, or scouring at the base of the stakes affecting the manual stake measurements. Throughout the study the heights of the OGREs were occasionally manually measured from the antenna ground plane to the snow surface; the mean bias was $2.5 \pm 3.1$ cm (n = 31), which is

the same sign as the bias found by Siegfried et al. (2017). This means the OGREs overestimate $H_r$ relative to the manual point measurements.

We also observe that the residuals are not uniform throughout the study, appearing higher in late fall and winter time months. This seasonal effect appears to correspond to periods of higher uncertainty, indicative of a rougher surface, or perhaps the variability is in part to scouring and pitting around the bamboo stakes. During the first year of the study, the temporal

offsets between manual stake measurements and OGRE measurements were also the largest. As for the OGRE, surface frost may not be as reflective, while these variations warrant a more robust investigation of a seasonal pattern in biases. Nonetheless, we observe a better overall precision than previously published results from the cryosphere, and a comparable bias relative to other studies.

### 3.3 Intercomparison of L1 and L2 Frequencies in Surface Determination

An additional strength of the OGRE is its multi-constellation and multi-frequency tracking ability, which results in hundreds of successfully calculated $H_r$ measurements for a 24 h period. It is important to analyze whether there is a statistically significant difference between L1 and L2 frequencies that arises due to different penetration depths or antenna phase center locations, which will influence how we consider these two signals when we aggregate our data.

We identify any individual satellite arcs across all constellations where both a L1 and L2 frequency from the given satellite

produce $H_r$ estimates in our processing routine. We then compute the mean of the biases between all these pairwise points by constellation (Table 1). Across all aggregated data, we find no statistically significant evidence ($p < 0.01$) that the biases differ from zero for any constellation. This suggests that the differences in frequency or antenna phase center location are not enough for a detectable difference over other sources of processing error or noise. These biases are smaller than those reported





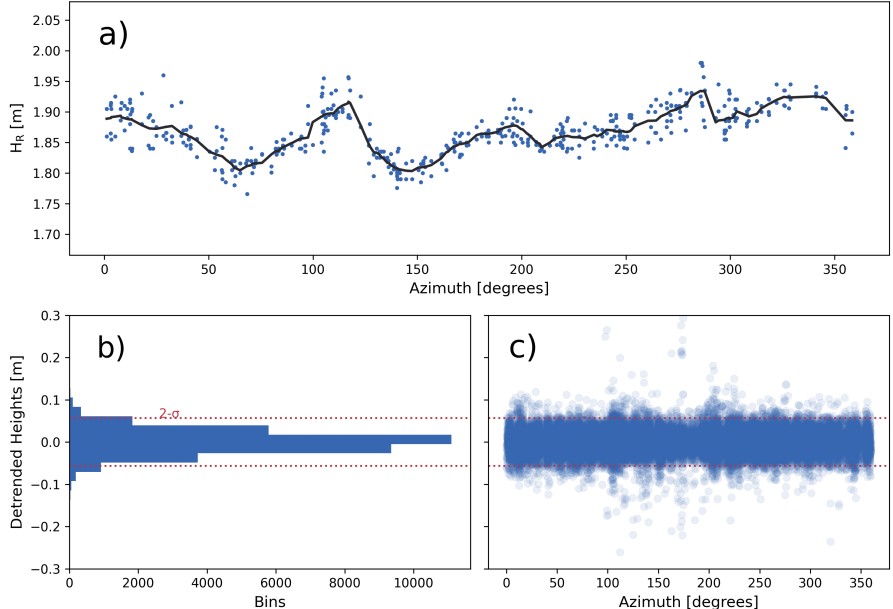

**Figure 4.** (a) Example 24 h period of $H_r$ measurements from an OGRE with the mean surface determined by passing the data through a kernel filter tuned to encompass 10 degree azimuthal windows. (b & c) All detrended surface height $H_r$ data points for the Bamboo Forest OGRE across all measurement periods are aggregated together and show an approximately normal distribution. The 1-$\sigma$ value (2.8 cm) is taken to be the precision value for any single $H_r$ estimate.

in Larson and Small (2016) for GPS signals, while these results allow us to further assess the precision of an individual $H_r$
estimate by considering L1 and L2 $H_r$ estimates together.

### 3.4   Individual $H_r$ Measurement Precision

Section 3.3 demonstrated that we can consider $H_r$ estimates from L1 and L2 together. As this study seeks to evaluate surface heterogeneities from $H_r$ estimates, we need to first understand the precision of individual $H_r$ estimates. In general, if we wish to assess the noise and precision of single $H_r$ estimates corresponding to particular satellite arcs, we would conduct a repeatability
test from one day to the next with an unchanging surface. However, while we consider a 24 h period of surface heights to be static, we cannot necessarily make the same assumption from one measurement period to the next. Given the large amount of data and the general agreement between L1 and L2 height estimates, we instead assess the variability about the mean surface for any period.

In Fig. 4a, we see an example aggregate of $H_r$ estimates for a 24 h period, plotted based on azimuthal location about the
OGRE. These $H_r$ estimates show a low frequency autocorrelation when plotted azimuthally. As $H_r$ estimates are semi-randomly sampled temporally, changes in atmospheric or surficial conditions throughout the 24 hour window cannot contribute to this low frequency signal. In the next section we discuss how these variations are linked to surface features and topography, but





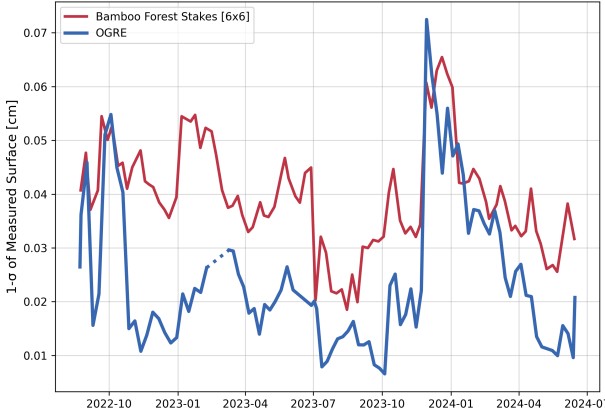

**Figure 5.** Bamboo Forest surface roughness (derived from 6 by 6 inner stakes) compared to OGRE surface roughness. Note that between 2023–02 and 2023–03, the OGRE did not record any data as it was temporarily removed from the field for a firmware upgrade and delayed in its redeployment.

in order to assess precision here we remove this azimuthal trend in the data by applying a fixed-bandwidth Gaussian kernel smoother to find the mean surface by azimuth. First, any outliers ($> 3\sigma$) are removed for a given 24 h data period. We then remove any days where there is a temporal trend in the data ($p < 0.01$), indicating that detectable accumulation occurred during the 24 h period. This occurred during 8 days throughout the study. The removal of outliers and temporal-trending days reduces the total number of $H_r$ points from 39,480 to 35,458. Finally, we define a bandwidth window that encompasses 10° azimuthal chunks of the data in order to apply the filter to remove the low frequency signal.

Aggregating all detrended measurement data yields a 1-$\sigma$ of 2.8 cm (n = 35,823) (Fig. 4). This value represents the precision of any single $H_r$ measurement, and may still encompass errors from varying surface conditions throughout the study and processing errors, however this value agrees with estimates made during past studies showing that individual satellite signals produce height estimates varying on the order of 2–3 cm (Larson and Small, 2016). Furthermore, Gutmann et al. (2012) estimates the formal error of the GNSS-IR method to be 2 cm, while Larson and Nievinski (2013) estimate this error to be 2.5 cm, which indicates these formal and processing errors largely constitute the estimate of this precision.

## 4 Surface Roughness Measurement Validation

### 4.1 A new technique for measuring surface roughness

We now evaluate the sensitivity of OGRE reflectometry-derived $H_r$ estimates to surficial features such as sustragi and dunes. For a given 24 h period, we first remove outliers as in Section 3.4. We then apply the same 10 degree fixed bandwidth kernel filter to derive the underlying low frequency signal, resampling the signal to 1 degree bins to ensure an even spatial sampling. Finally, we fit a sinusoid with a period of 1 per 360° to the data in order to remove the underlying surface slope, and we take



the standard deviation of the detrended data. This value is taken as the OGRE-derived surface roughness. Similarly, for the Bamboo Forest data, we remove the plane of best fit from the subset of 36 stakes closest to the OGRE and take the standard deviation to represent the surface roughness, covering a spatial extent of 40 m by 40 m. Over the study period, we find that the OGRE-derived surface roughness correlates with the Bamboo Forest stakes (r = 0.74, p = $1.5 \times 10^{-14}$, n = 78), again using a $\pm 48$

h search for comparisons (Fig. 5). During the final year of the study, where the prominent heightened surface state roughness is detected by both methods, the correlation is 0.89. We also conducted this comparison for increasingly larger and smaller subsets of bamboo stakes, with monotonically decreasing correlation coefficients from 0.74 for the 36 stakes to 0.56 for the entire 121 stake network, and a correlation of 0.51 for the inner four stakes. To test for different scales of roughness, we also varied the bandwidth window of the kernel filter from 5 to 40 degrees, and found a similar results, suggesting the roughness

features at different scales are correlated.

The 36 stakes, which are spaced at 8 m in a grid, act like a lowpass filter and are sensitive to surface features that have wavelengths close to or exceeding their spacing distance. Meanwhile, large-scale features with a wavelength greater than the distance of the spatial extent (40 m in either direction) are removed when the plane of best fit is removed. Meanwhile, the OGRE data is sampled circularly and for the range of antenna heights in this study, and this comparison indicates that the

technique is most sensitive to the surface 4–20 m radially from the OGRE location. Each $H_r$ estimate is effectively a spatially averaged surface based on the Fresnel geometry, and thus features with wavelengths smaller than these ellipse widths (2–4 m) will minimally bias the height estimates. We can conclude that the OGREs in this configuration are sensitive to roughness features on the order of several-meter wavelengths. In fact, both the OGRE and Bamboo roughness estimates are on the same order scale as the sastrugi amplitudes measured at Summit by Albert and Hawley (2002). While we cannot explicitly connect

these undulations to physical locations and the spatial footprints retain some ambiguity, we can nonetheless clearly detect heightened or attenuated roughness states within the meter-scale sensitivity of the OGREs, especially when our roughness estimates exceed the 2.8 cm individual $H_r$ precision previously calculated.

Perhaps the largest source of error between the two roughness estimates is the spatial extent: while surface roughness can be correlated across spatial scales due to its fractal nature, the differences in footprint size between the OGRE and Bamboo

Forest will lead to different measurements (Zuhr et al., 2021). We also made the assumption that no accumulation takes place during the 24 h measurement periods; however this is not always the case, as eight periods had a statistically significant slope in $H_r$ versus time (p < 0.01) that indicated a sub-daily surface change rate of several centimeters, with the same sign as the week-over-week change surrounding that day. Furthermore, the standard deviation calculation is sensitive to outliers, and while we remove measurement outliers for any given survey greater than $3\sigma$ for the stakes, error propagation and systematic errors

could compound, especially on dates far from the auto level survey conducted during June 2024. As previously discussed, the OGRE footprint shrinks throughout the two year study as the surface accumulates relative to the antenna, which leads to further mismatch between the two footprints. However, the results presented thus far provide a compelling demonstration that most of the azimuthal variability in $H_r$ measurements is due to real, time-varying structural features in the sensing domain.

We further validate the extent and properties of the sensing area by examining the 10 cm resolution TLS surface scan

(Fig. 2a) in the Bamboo Forest from 16 June 2024, where we found only a moderate relationship (r = 0.36) between OGRE $H_r$




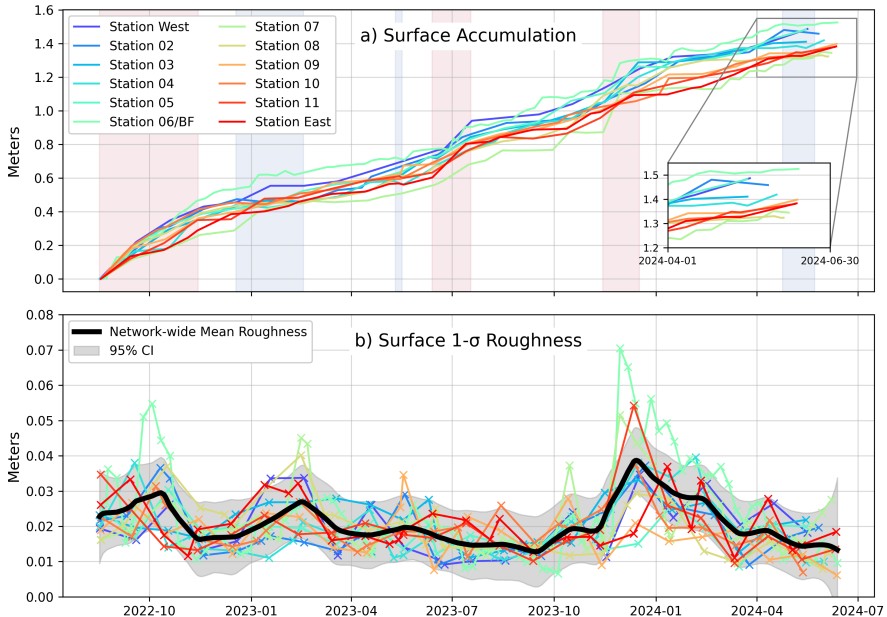

**Figure 6.** (a) accumulation through time of all twelve OGRE stations, with periods of elevated accumulation or low/negative accumulation highlighted in light red and blue, respectively. The OGRE station time series are colored from blue to red to indicate location from west to east. (b) 1-$\sigma$ (roughness) values through time of all stations, with the mean network-wide value indicated by the thicker black curve.

estimates mapped to the TLS surface, with the slopes removed. By imposing Fresnel zones calculated from 0.1 degree elevation angle spacing from 5° to 25° on the TLS surface, we can reconstruct the GNSS-IR footprint to extract mean surface values. A discussion on the potential sources of error that lead to the lower agreement between the datasets is discussed in Appendix B.

## 5 Results and Discussion

### 5.1 Network-Wide Spatial and Temporal Patterns in Accumulation

Incorporating data from all twelve stations, we show accumulation and surface roughness based on the mean and standard deviation of 24 h intervals of recorded data, respectively, at each station throughout the study period (Fig. 6). From August until November 2022, accumulation rates were nearly ten standard deviations higher than the overall mean rates for most stations. Notably, the annual accumulation rate from September 2022 to September 2023 is nearly 95 cm y$^{-1}$ at the Bamboo Forest OGRE, which is much higher than rates previously found by Dibb and Fahnestock (2004) (64–65 cm$^{-1}$), and in the upper 95th percentile of the $71 \pm 11$ cm y$^{-1}$ mean rate found over ten years of Bamboo Forest data (Castellani et al., 2015). While much of Greenland experienced melt due to several atmospheric river events during September 2022 (https://nsidc.org/ice-sheets-



today/analyses/record-september-greenland-ice-sheet-melt), this precipitation fell as snow at Summit Station and contributed to the anomalously high rates of accumulation over the period.

In 2023, several stations again experienced local accumulation rates much higher than their averages in June and July and from mid-October to December, while 2024 featured no such enhanced accumulation prior to the end of the study. Meanwhile, near-zero or negative accumulation rates were observed for a number of stations between mid-December 2022 and March 2023, and mid-April to June 2024. Temporal patterns of accumulation suggest that the observations made by Dibb and Fahnestock (2004), Howat (2022), and Castellani et al. (2015) still hold true: there is a marked increase in accumulation for all stations

during the late summer and early fall when precipitation is generally high but firn compaction rates decrease from summertime highs, and a lower period of accumulation (or negative accumulation) in the late winter and spring, which is perhaps driven by enhanced negative processes described in Eq. 1. However, the Bamboo Forest OGRE exhibited a mean annual rate of 77.8 $\pm$ 10 cm y$^{-1}$, which trends higher than previously reported annual rates for this location. The Bamboo Forest OGRE recorded an especially high rate of accumulation in the winter of 2023 relative to the other stations, and we posit that a combination of

winds and its proximity to drift-inducing structures may have led to the emphasized surface height increase.

      From Station 11 (closest to the divide) across-slope to Station West, the slope is 0.067° and spans 34.7 km with a vertical drop of 43.7 m, and these stations exhibit an across-slope difference in accumulation over the two year period, which is visible in the inset in Fig. 6. We find the mean rate of accumulation over the study period by averaging a sliding window of 365 days; Station 11 has a mean accumulation rate of 70.9 $\pm$ 4.2 cm y$^{-1}$ and Station West exhibited a mean rate of 77.4 $\pm$ 4.7 cm y$^{-1}$.

Because the uncertainties for each station are correlated, we use a T-test of the slopes of a linear regression fit over the entire period of data to confirm statistical significance in the accumulation rates between these two stations (p = 0.04). Furthermore, we find that MERRA-2 precipitation rates vary between these two locations by a similar amount: 79.4 $\pm$ 4.9 cm y$^{-1}$ and 85.9 $\pm$ 5.1 cm y$^{-1}$, using an assumed density of 300 kg m$^{-3}$ to convert liquid volume to estimated snow thickness, a common value of surface snow at Summit (Montgomery et al., 2018). Here, the difference is 6.5 cm y$^{-1}$, which mirrors the 6.5 cm y$^{-1}$ difference

of the OGREs. This indicates that both MERRA-2 and the OGREs are sensitive to the dominant variations in moisture that originate from the west and southwest in this region (Bolzan and Strobel, 1994).

      The across-network spatial variability in accumulation takes two factors into account: semi-random localized accumulation within the spatial footprint of each OGRE, and a small along-slope increase in precipitation from the divide to the west. Since each station often logs on independent days of the month compared to the others, we interpolated $H_r$ measurements

to daily measurements, removed the station-specific accumulation trend, and calculated the standard deviation for each day between stations to find the inter-station variability in accumulation driven by localized differences in Eq. 1. The mean standard deviation between the detrended data is 2.6 cm, suggesting that on monthly scales and at 10 km spacing within our study area spatial variability in accumulation is small between stations. However, we note that from November to January 2024, the mean standard deviation of the detrended network peaks at 4.0 cm, which is the same period of heightened local variability for many

stations (Fig. 6).





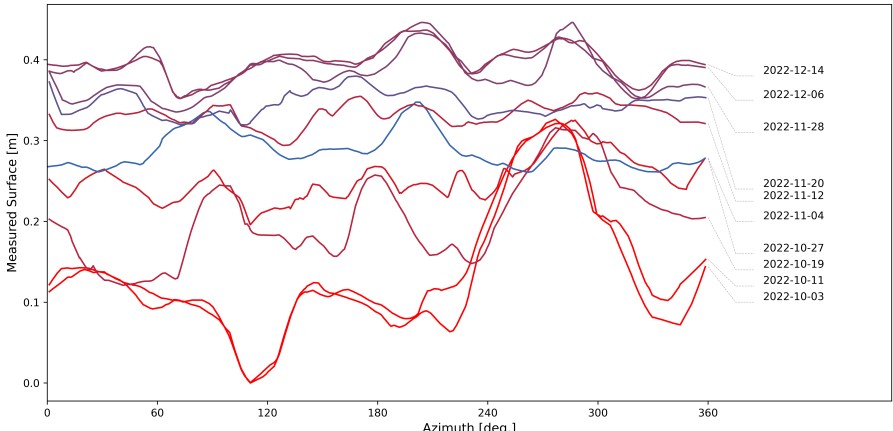

**Figure 7.** Several GNSS-IR-derived surfaces from the Bamboo Forest OGRE plotted azimuthally, showing patterns of accumulation between October and December 2022. Color is based on the correlation of each data period to the first profile on 03 October 2022, with red indicating the most correlation and blue indicating the least correlation.

## 5.2 Network-Wide Temporal Patterns in Surface Roughness

The network of OGREs also exhibits periods of elevated surface roughness values by several or a majority of stations. There are three periods when a majority of stations show heightened roughness above the 2.8 cm threshold: in September and October of 2022, February and March of 2023, and December of 2023 through February 2024. These periods indicate a seasonal pattern of
heightened roughness that corresponds to either higher snowfall that can be heterogeneously distributed or eroded, or periods where winds play a dominant part in snow removal with negative or little accumulation occuring. Interestingly, no heightened roughness is detected in the summertime months despite distinct periods of comparable, extended high or low accumulation events to winter months. The drivers of accumulation in Eq. 1, including wind, latent heat flux, and compaction are all seasonal (Castellani et al., 2015), and while they only account for a small percentage of the total accumulation in the Summit region
(e.g. Lenaerts et al., 2012), these factors may play a larger role in driving the seasonality in roughness observed here.

The agreement between most stations indicates that the snow features which are large enough in amplitude to be detected must also have a spacing wavelength that falls within the bounds of the footprint. Albert and Hawley (2002) noted that the wavelength between snow features showed little seasonal pattern and spanned 5–20 m. If the wavelength did vary spatially on these scales during any measurement period by the OGREs, then this could partially explain the variability in roughness
measurements between stations.

## 5.3 Local Spatial and Temporal Patterns in Surface Roughness

Figure 7 shows several OGRE-derived $H_r$ profile estimates, plotted based on azimuthal direction. For ease of viewing, we run a local polynomial filter over each data series to provide a continuous line for easy viewing while masking individual





$H_r$ estimates. We also colored each time series based on its correlation to the first (03 October 2022) measurement, with red
indicating a correlation coefficient of 1 and blue indicating the lowest correlation coefficient value. Here, the surface starts
with a heightened roughness state, with a large peak spanning the western quadrant of the instrument. Through time, the
surface evolves due to positive or negative accumulation processes (Eq. 1). We prescribe a physical explanation of the surface
evolution: the surface generally increases, but not uniformly. As the troughs around the 30 cm October 03 feature fill in, the
peak remains at the same elevation or even erodes until it is completely buried. This pattern matches that described by Filhol
et al. Filhol and Sturm (2015), whereby wind affected snow will show preferential erosion or deposition based on exposure:
wind sheltered regions fill in first while wind exposed features remain exposed to erosion.

Furthermore, we note that the correlation between each subsequent time series and the first measurement is not monotonically
decreasing; this may suggest that well-sintered features may be preserved in the snowpack after an accumulation event and
become re-excavated, or play some part in the overlying snow topography (Zuhr et al., 2021). Ultimately, care must be taken
in interpreting these daily time series given the geometric constraints discussed in Section 2.4. For example, the 30 cm "peak"
on October 03 is a shallow mound that spans over a 100° field of view from the perspective of the instrument. As each reflector
height estimate is composed of inputs from a variety of satellite elevation angles and therefore varying Fresnel zone size and
location, these apparent features may dampen or spatially average any true physical feature, or preferentially emphasize raised
surfaces due to radio shadowing (Bourlier et al., 2006).

**5.4    Surface Roughness and Accumulation Connection to Meteorology**

For a process-level understanding of the drivers of accumulation and surface roughness expressions observed by GNSS-IR
and illustrated in the previous section, we compare OGRE data to MERRA-2 precipitation and winds (Fig. 8). We specifically
examine the data from the Bamboo Forest OGRE, given the weekly sampling rate. For each period between samples, we derive
interval precipitation by calculating the cumulative precipitation scaled to a snow thickness using an assumed density of 300 kg
m$^{-3}$, and we derive interval winds by calculating the median wind speeds during each period. We then compare these MERRA-
2-derived variables to accumulation ($\Delta H_r$) from the OGRE, along with surface roughness change, $\Delta\sigma$. This comparison is
similar to those made by Picard et al. (2019) in Antarctica and Zuhr et al. (2021) in Greenland.

MERRA-2 precipitation and MERRA-2 winds have a moderate correlation and high statistical significance, which may be
explained by observations by Pettersen et al. (2018) that the ice and mixed phase clouds that bring moisture to the Summit
originate from strong southerly storms that exceed average yearly wind speeds. Meanwhile, OGRE accumulation and MERRA-
2 winds show low correlation and no statistical significance; this may be in part due to negative OGRE accumulations that are
a result of more erodible, soft snow conditions (Filhol and Sturm, 2015). When we compare net OGRE accumulation with
winds or remove the negative accumulation values, we gain statistical significance, which may hint at a bimodal accumulation
process linked to high winds.

Unlike in Picard et al. (2019) or Zuhr et al. (2021), we correlate the surface roughness change from one period to the next as
opposed to the measured surface roughness because of the interval of our measurements. For example, a surface may maintain a
rough state for several days if no new snow or winds occur, and therefore these two variables are not as comparable, whereas the





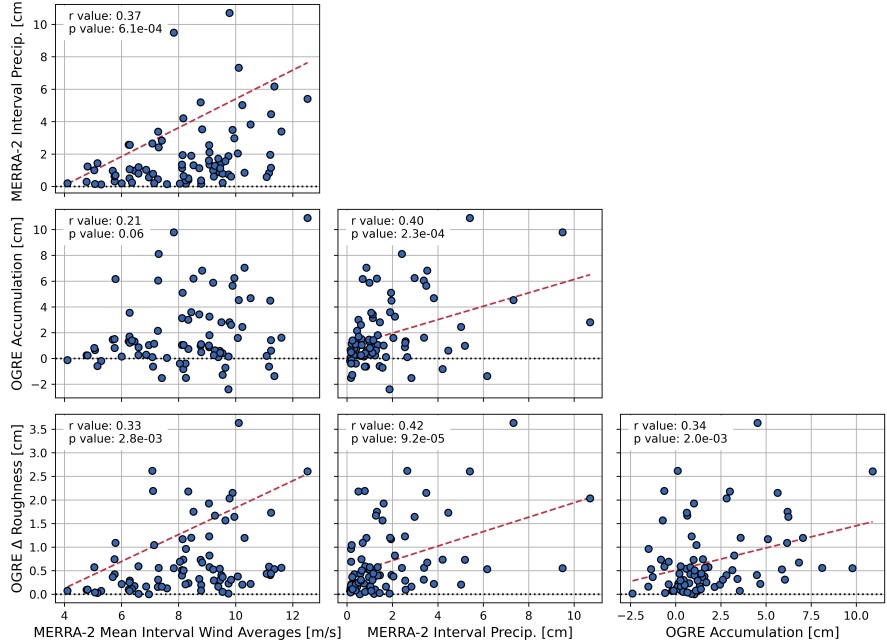

**Figure 8.** Intercomparison between weekly Bamboo Forest OGRE accumulation and roughness change data with MERRA-2 winds and precipitation.

change in surface roughness state can physically be explained by the presence or lack of precipitation and winds. The moderate correlation and high statistical significance between surface roughness change and winds suggests that wind speed can play a

role in surface roughness, but other factors such as the hardness of the existing snow must be considered, alongside cumulative precipitation and surface accumulation, which also have moderate a correlation when compared to surface roughness change. Overall, the interpretation that higher winds correlate with increases in surface roughness is supported by the observed behavior of wind-driven surface roughness observed at the South Pole (McConnell et al., 1997), but our study only indicates whether winds directly lead to magnitude change in surface roughness and not whether the surface became more or less rough compared

to its prior state.

Finally, OGRE accumulation and cumulative precipitation show a moderate correlation with high statistical significance. The relationship between precipitation and OGRE accumulation is skewed downwards from a 1:1 relationship, suggesting that other factors, such as sublimation, compaction, and variability in surface density may need to be included in a more comprehensive analysis, especially as MERRA-2 precipitation alone cannot physically explain negative accumulation values.



## 6 Conclusions

We have expanded the scope of the GNSS-IR technique in the cryosphere by validating GNSS-IR-derived measurements, examining the spatial extent and temporal reliability of this technique, and assessing its sensitivity to local surface heterogeneities. Here, averaged 24 h periods of GNSS-IR estimates are biased by 2.7 cm in low angle, dry snow, and precise to 3.0 cm. This precision is better than any previously reported in the literature, and this is in part due to a comparison to a more spatially representative validation dataset. However, the biases between the GNSS-IR heights and the validation dataset warrant further investigation as to the slight differences in wintertime and summertime. It is now common for receivers to track both the L1 and L2 signals of multiple constellations, and we also show that there are no biases between the two signal types, allowing us to effectively double the size of our dataset, while individual measurements show a precision of 2.8 cm.

We also assessed the spatial extent of this surface-sensing technique. While we model the moving ground reflection locations of each satellite signal arc and find them to span from 4–35 m radially for a 2 m antenna, we found that the derived surface height estimates across the 1–2 m antenna height range are most sensitive to the inner 4–20 m about the instrument, based on the best correlation with a subset of snow stakes. The characteristics of this sensitivity have several implications for the sensing footprint. First, this technique can average over a larger area than point measurements and remove high-frequency surface noise, providing a more spatially representative measurement for daily or sub-daily surface accumulation. Second, this technique provides additional context for understanding the circumferential spatial heterogeneity that we estimate, limiting our window of sensitivity to centimeter- to meter-sized features that span multiple meters horizontally. We find that these surface roughness values of the OGRE share the same magnitude and pattern of roughness variations with a 36 stake network grid spaced at 8 m (r = 0.74). Temporal sampling offsets, differing spatial footprints, and measurement error from the validation dataset likely account for disagreements in roughness magnitude and timing between the two datasets, but they both show fidelity in detecting increased roughness states.

We then assessed accumulation and surface roughness values derived from a network of twelve low-cost GNSS stations in the interior, dry snow zone region of Greenland for spatial and temporal patterns compared to historical data. Accumulation largely consistent with historically-observed late summer and early fall enhanced height increases, along with decreased rates in the springtime. This network also detected a centimeter-level cross-slope variation in accumulation at yearly scales, consistent with expected higher downslope accumulation. Meanwhile, surface roughness trends exhibited several wintertime periods of increased roughness, and it was during these periods that the greatest spatial variability (±4.0 cm) was observed in accumulation between stations. Furthermore, these results suggested that both erosional feature formation and deposition heterogeneities can drive an increased roughness at cm to m scales in the Summit vicinity.

On a process level, we demonstrated the utility of GNSS-IR measurements for better understanding snowpack stratigraphy and the preservation or erosion of snow layers through time. We connected height and roughness measurements with wind and precipitation from MERRA-2 to show that these synoptic variables play a role in the accumulation and structure of snow in the Summit vicinity, but also highlight that there must be other processes or variables, such as sintering time, that may play a more dominant role in the preservation or erosion of snowpack. This exercise highlights the GNSS-IR technique as an easily



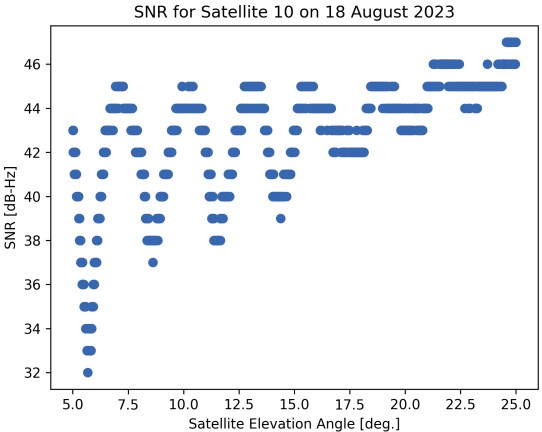

**Figure A1.** Example SNR data from a single rising satellite arc from an OGRE on 18 August 2023, showing the characteristic integer binning and degraded interferometric quality at higher elevation angles.

executable, adaptable method that can shed light on processes that have previously been difficult to measure in the field, such as

wind scour and firn compaction. The ease from which these GNSS instruments can be configured to take daily measurements further accentuates the ability of this technique to illuminate short-timescale processes and trends.

Ultimately, these results are local to a relatively flat, accumulating 30 km region in the vicinity of the Greenland Ice Sheet summit, but the methods presented here show promise throughout the cryosphere. Low cost GNSS devices such as those presented in this paper are more easily deployable in larger quantities due to their smaller size and weight, cost, and energy

efficiency, all while providing high-quality results and requiring no special configuration other than an elevated antenna. The reliability and resolution of the GNSS-IR technique is demonstrated to be sensitive to low accumulation signals, while we also show for the first time that the variability in individual $H_r$ estimates is due in large part to surface roughness. These measurements can be combined to better assess ice core stratigraphy, measure turbulent heat flux in real time and detect daily changes driven by precipitation, compaction, wind, and sublimation, or serve as ground validation measurements for a variety

of remotely sensed variables.

*Code and data availability.* RINEX files and metadata from the OGRENet array are provided at https://doi.org/10.18739/A2736M41C (Pickell and Hawley, 2024a). Design files and firmware for the open-source OGRE instruments are available at github.com/glaciology/OGRE. Bamboo Forest data can be downloaded at https://conus.summitcamp.org/mirror/summit/ftp/science/bamboo_forest/. GNSS-IR data were processed at https://doi.org/10.5281/zenodo.10796409 (Larson, 2024).



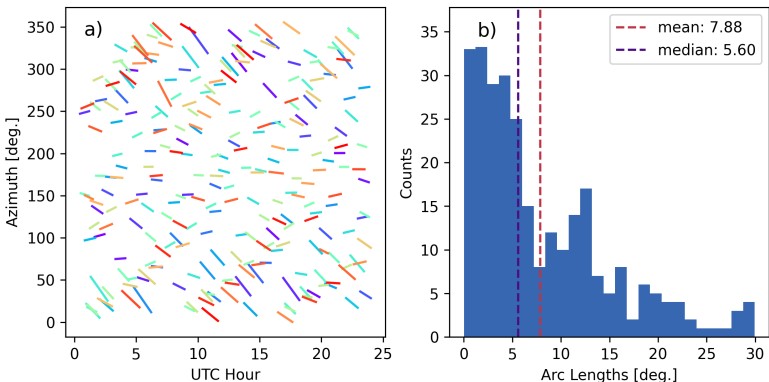

**Figure A2.** (a) Sample 24 h period of satellite azimuth projections, with each satellite represented by a different line, showing that every satellite has some azimuthal variation as it rises or sets, relative to the OGRE. (b) The mean arc is 7.88°, and the maximum is 29.95°. Across the entire study period, the mean arc is 8° and the maximum arc is 40°. In this study, we take the mean azimuthal value for each arc to represent the azimuthal location of that satellite arc.

## Appendix A: Further discussion on ground footprint of GNSS-IR

Nievinski and Larson (2014) show that for a 2 m antenna height, there is an uneven weighting for the sensing footprint of reflectometry: the peak region of importance lies somewhere between 13 and 15 m (corresponding to about 10 degree satellite elevation angle), with regions closer to the antenna defined by a shorter tail (less importance) and regions farther than the peak characterized by a more gradually decreasing tail of importance. This spatial pattern arises for several reasons. At low satellite elevation angles, corresponding to a reflection farther from the instrument, the interferometric SNR pattern is often less noisy than the nearby portion of the signal (Fig. A1). Of course, this is further complicated by the ability of the Lomb-Scargle to determine the peak frequency, and perhaps by the integer-binning of SNR readings by the receiver, which is a characteristic of the low-cost chip in the OGRE instruments.

Nonetheless, this footprint interpretation is somewhat substantiated by the main text, as the 13–15 m dominant region falls within the 4–20 m zone identified as the most important region. However, it is unclear whether the stronger weighting beyond 15 m, and quickly diminishing weighting closer than 13 m can be confirmed. In fact, our comparisons with both the TLS surface data and correlations to individual bamboo stakes show that nearby surfaces are in fact more represented by the GNSS-IR solutions.

A second point of note is that each $H_r$ footprint corresponding to each arc is not necessarily perfectly radial to the instrument, as illustrated in Fig. 2. Each satellite arc contains an azimuthal component throughout the sky: for each 24 h period at an OGRE station at Summit, we measured the azimuthal angular distances, projected to the horizontal horizon plane, that the observed satellites travel as they rise or set through the critical elevation angle range of 5° to 25°. The mean azimuthal travel from all data is 8°, and the maximum azimuthal travel was found to be 40° (Fig. A2). This means that there will be some azimuthal error



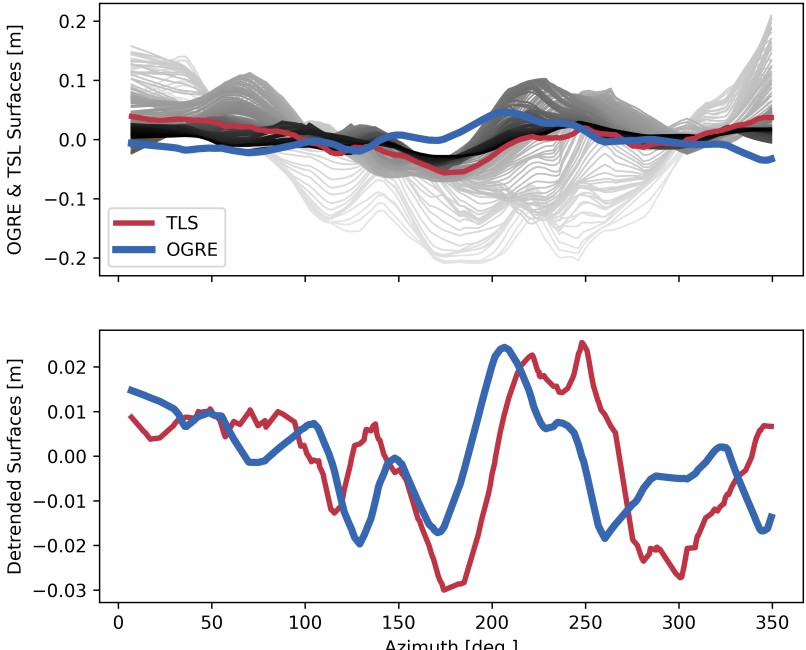

**Figure B1.** From the 16 June 2024 TLS survey of the Bamboo Forest, we compare the OGRE height estimates to the TLS surface. The TLS surface is the composite of the sampled surfaces corresponding to the reflection points of simulated GNSS-IR reflected signals from R = 5° (light gray) to R = 25° (black).

in positioning singular reflector heights, which in turn will induce an error on estimating averages in particular quadrants, or
applying a fixed-bandwidth Gaussian kernel filtering technique. Furthermore, there is an azimuthal-dependence for the length of the azimuthal arc; satellites to the north tend to move a greater horizontal distance across the sky than those to the south. This complicates any attempt to specifically pinpoint the location of a $H_r$ estimate, and care must be exercised when attempting to directly compare a $H_r$ estimate to a specifically located measurement such as at an individual snow stake.

## Appendix B: TLS Comparison and Discussion

On 16 June 2024, we used a TLS to create a 10 cm resolution DEM of the Bamboo Forest surface. The OGRE concurrently recorded 24 h of data during this period. For each $H_r$ estimate, defined by its azimuthal location, we sample the TLS surface based on the centroid of the evolving ellipse defined by the first Fresnel zone (FFZ) as it migrates closer to the OGRE and shrinks in size. Figure 2 demonstrates this geometry, which is driven by the satellite elevation angle. The azimuthal TLS height estimates are aggregated to create the TLS surface topography estimate for comparison to the OGRE surface (Fig. B1). A



sample of TLS-derived surfaces for the annular distances defined by the 5°–25° range are shown from light to dark gray, respectively. These FFZ annuli are averaged together to create the overall TLS-derived surface estimate, which we believe to be a good sampling approximation of the FFZ geometry.

The two surfaces are adjusted so their mean value is centered at zero, yet and the variance of the residuals between the OGRE and the TLS surfaces is higher than if we simply compared the OGRE surface to a planar surface. This low agreement

could be attributed to a number of factors, including errors from the footprint positioning assumptions discussed in the previous section, errors from a slightly tilted antenna, and a slight evolution of the snow surface during the 24 h OGRE data period. Furthermore, this comparison averages across the full radial distance defined by the 5°–25° elevation angles (per the FFZ pattern), but perhaps the GNSS-IR surface is more sensitive to certain regions. For instance, the example SNR OGRE data shows a cleaner sinusoid at low satellite elevation angles and therefore may be more heavily weighted towards these angles in

the frequency estimation, and thus the corresponding surface further from the antenna is more important. Conversely, the main text argues that the GNSS-IR derived measurements more closely agree to point measurements near the OGRE, which would favor the data from the 15°–25° elevation bands.

To investigate these differences, we remove the area-wide surface slope from both the OGRE and TLS data and compare the resultant surface estimates from both techniques (shown in the second plot in Fig. B1). Here, the surfaces are no more

correlated than previously (r = 0.36), however they visually share peaks and troughs, albeit somewhat offset, especially outside the 150°–200° range. The non-linear phase shift may be indicative of the effect demonstrated in Fig. A2, where more northerly $H_r$ estimates are in fact biased in their locations based on the arcing path of each satellite. The similarity in magnitude of the peaks could explain how the OGRE, while showing low agreement with TLS surface measurements, is still sensitive to the detrended surface roughness measurements described in this text.

*Author contributions.* DJP designed the study and DJP and RLH carried it out. DJP processed the GNSS-IR data and performed the data analyses. AL generated the TLS digital terrain model. DJP prepared the manuscript with contributions from all co-authors.

*Competing interests.* The authors declare that they have no conflict of interest.

*Acknowledgements.* We thank the 2022–2024 OGRENet field team members for instrument deployment data collection assistance, including Jamie Good, Olivia Moehl, and Joel Wilner. We also thank Battelle Arctic Research Operations for logistics support, and especially the

dedicated team at Summit Station, along with the science technicians who gathered weekly stake height measurements. Finally, we thank the Cold Regions Research and Engineering Laboratory instrumentation team for the TLS support. This project was funded by the US National Science Foundation Office of Polar Programs grant #2028421



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
