# Peer review of "Spatio-temporal patterns of accumulation and surface roughness in interior Greenland with a GNSS-IR network"

_EGUsphere, 2024_

## Author Response (AR1)

**Response to Reviewer #1**

Derek Pickell et al.

We thank the reviewer for their helpful and constructive comments. Our responses to the italicized text are recorded below. Changes implemented in the latest version of the manuscript are in red.

*Nice and very extensive analysis done on a set of low cost GNSS stations near Summit Station in Greenland. The authors compare their snow accumulation measurements derived from GNSS-IR to a set of snow stakes and find only small differences between the two datasets. The authors also attempt to quantify surface roughness within the sensing footprint and find a moderate correlation.*

*I don't have many comments though I wonder if additional details could be included in Figure 8--specifically, if for the precip panels the points could be colored by wind speed, and vice versa. Does that help at all with understanding some of the source for the scatter?*

Response: This suggestion is implemented in the proposed Figure 8 below. This may be helpful in interpreting some of the outlier points; for instance, it is now easier to see that the largest change in surface roughness is accompanied by both high accumulation and high winds. However, our overall analysis does not change as the low correlations for each comparison make it hard to attribute the scatter to solely winds or precipitation, and this leaves room for a future, in-depth analysis of additional variables such as time-varying compaction rates and sintering.

[Figure]

**Figure 8**. Intercomparison between weekly Bamboo Forest OGRE accumulation and roughness change data with MERRA-2 winds and precipitation.

With this updated figure, the following changes will also be made:
- Changed wind x-axis label to 'MERRA-2 Mean Interval Winds [m/s]'
- Calculation of roughness: a small bug meant that the algorithm to estimate GNSS-IR-derived roughness and roughness change did not exactly mirror that described in the methods section. This was updated accordingly, reflecting the small shifts in correlations in the bottom row.
- An additional 3 days of data from June 2024 were added to the above figure, following the publishing of new MERRA-2 reanalysis data to mirror the full dataset.

With the data changes presented here, we note that the Accumulation - Roughness correlation is now the highest, and that the Accumulation - Winds has become statistically significant at $p < 0.05$. The text will be updated to capture these changes. We may also add a sentence in our analysis that details the correlation values between surface roughness (not surface roughness change) and MERRA-2 winds/precipitation and OGRE accumulation, as we find that higher surface roughness correlates with median winds ($r = 0.41$) with high statistical significance, while there is very low correlation with accumulation or precipitation. This suggests that accumulation, which is indicative of the net gain/loss of material from one period to the next, provides the necessary material to modify the roughness state (either by surface smoothing or by irregular deposition), but higher surface roughnesses more often correspond with periods of high winds, such that winds drive irregular deposition or enhanced erosion.

Figure 8: updated to the proposed figure above. Associated changes in the text include:
Line 403: changed "median" to "mean"
Line 405: changed "moderate" to "low-moderate"
Line 408: changed "low correlation and no…" to "a lower correlation and marginal…"
Line 409: added "in winds"
Line 411: changed to "increase statistical significance and the correlation value to 0.27, but we lack the necessary data to investigate a potential…"
Line 421: added "These variables have low to moderate correlations when compared to surface roughness change, with positive precipitation or accumulation perhaps providing the necessary material to modify the roughness state either by surface smoothing or by irregular deposition."
Line 430: added "...nor can the smaller OGRE spatial representativeness exactly capture the MERRA-2 interpolated estimates."

*Also, do you have any ideas for why your analysis showed minimal differences between L1 and L2 signals, whereas (as you note) the Larson research group found more substantial differences?*

Response: We suspect that this may be due to improvements in the signal correlation technology on the more modern (albeit low-cost) u-blox receiver chip running on the OGRE. Larson et al. 2010 note that the geodetic receivers used in their study are better at cross-correlating L2C signals compared to the older L1 C/A signal, which has a shorter chip rate and is more prone to noise. In this study, we do not observe any significant difference in L1 and L2 SNR signal strength or noise characteristics across the 5-25 degree satellite elevation range, resulting in more comparable reflector height retrievals. Another factor that may lead to different results is the differences in antenna technology: for example, the ANN-MB-00 antenna used in this study has a ~1.0 mm difference in vertical phase offsets between L1 and L2[*], which is smaller than the difference in phase centers in some older antenna models.

Line 220 added: "…perhaps due to improvements in L1 tracking or a smaller phase center difference in the antennas used in this study"

*Finally, do you have any comment for what might happen to the reflection data if another melt event were to occur*

*at Summit, as I believe one did in 2019?*

Response: There was at least one above-freezing day during the course of this study, however the weekly to bi-monthly resolution of the OGREs do not allow us to isolate the effects of these sub-weekly events. We have now deployed continuously logging stations on-ice, and would expect that melt events that change the surface height on the order of centimeters will be detected by an OGRE. This paper focuses on reflectometry in the dry snow zone, and increased melt events in the region warrant further studies to investigate how or if changing snow facies (which change the dielectric/reflecting properties of the material) lead to different levels of precision or surface-property-dependent biases in height measurements.

No changes to manuscript in response to this comment.

Cited:
K. M. Larson, J. J. Braun, E. E. Small, V. Zavorotny, E. Gutmann, and A. Bilich, "GPS multipath and its relation to near-surface soil moisture," IEEE J. Sel. Topics. Appl. Earth Observ., vol. 3, no. 1, pp. 91–99, Mar. 2010, doi: 10.1109/JSTARS.2009.2033612.

*https://content.u-blox.com/sites/default/files/documents/ANN-MB_DataSheet_UBX-18049862.pdf

Response to Reviewer #2
Derek Pickell et al.

We thank the reviewer for their helpful and constructive comments. Our responses to the italicized text are recorded below:

*The authors have established and analyzed a network of low cost GNSS instruments to measure snow accumulation and surface roughness in the vicinity of the Summit Station in Greenland through the GNSS-IR technique. The use of GNSS technology for this type of measurement is very interesting as well as the use of low cost instruments. The experiment was well designed and conducted.*

*I have just a few minor remarks, concerning the presentation of the results.*

*No information about the type of low cost GNSS used*

Response: An appropriate place for more detailed information would be line 77 in Section 2.1, and we can include the GNSS chip model in addition to the patch antenna model.

Line 80: added "The OGRE is built on a u-blox ZED-F9P multi-band, multi-GNSS chip." and … "u-blox ANN-MB".

*In the Abstract: "show a statistically significant (p < 0.01)". What do you mean for "p"? To avoid any doubts it is better to define it*

Response: We have decided to remove the text "(p < .01)" in this sentence here, and will define it where appropriate (see line 194).

Line 14: (p<.01) removed.

*Line 192: (n=76), I suppose that "n" means "number of observations", isn't? So, I suggest to write (with 76 observations), later too.*

Response: We will include something like the following: "The dataset includes 76 observations (n = 76)."

We have opted not to implement this change as we find "n" and observations to be redundant during the first occurrence of "n" on line 197.

*Line 194: (p = 1.4 x10-9), please write the meaning of "p", that is used also later*

Response: We will include the following text prior to 'p': "…(p = 1.4x10e-9); where p represents the probability of observing these results under the null hypothesis."

Line 221: We have found the phrasing in this line to be sufficient: "Across all aggregated data, we find no statistically significant evidence (p < 0.01)."

*Line 295: "(64–65 cm-1)" ==> "(64–65 cm yr-1)"*

Response: This typographic error will be corrected.

Line 295: corrected.

Beyond the aforementioned corrections in red, the attached document includes grammatical and phrasing revisions. Here, we highlight the following (minor) numeric and wording corrections that influence the interpretation of results:

Figure 2 Caption: specified "L1 frequency" when describing the Fresnel zone.
Line 241: sample size corrected to 35,340.
Figure 4: corrected label to "Counts" instead of bins and re-plotted data correctly.
Line 260: sample size corrected to 76 from 78.
Line 414: added a missing negative sign to "-2.7"